# Cross-Reactive Antibodies to SARS-CoV-2 and MERS-CoV in Pre-COVID-19 Blood Samples from Sierra Leoneans

**DOI:** 10.3390/v13112325

**Published:** 2021-11-21

**Authors:** Rodrigo Borrega, Diana K. S. Nelson, Anatoliy P. Koval, Nell G. Bond, Megan L. Heinrich, Megan M. Rowland, Raju Lathigra, Duane J. Bush, Irina Aimukanova, Whitney N. Phinney, Sophia A. Koval, Andrew R. Hoffmann, Allison R. Smither, Antoinette R. Bell-Kareem, Lilia I. Melnik, Kaylynn J. Genemaras, Karissa Chao, Patricia Snarski, Alexandra B. Melton, Jaikin E. Harrell, Ashley A. Smira, Debra H. Elliott, Julie A. Rouelle, Gilberto Sabino-Santos, Arnaud C. Drouin, Mambu Momoh, John Demby Sandi, Augustine Goba, Robert J. Samuels, Lansana Kanneh, Michael Gbakie, Zoe L. Branco, Jeffrey G. Shaffer, John S. Schieffelin, James E. Robinson, Dahlene N. Fusco, Pardis C. Sabeti, Kristian G. Andersen, Donald S. Grant, Matthew L. Boisen, Luis M. Branco, Robert F. Garry

**Affiliations:** 1Zalgen Labs, LCC, Germantown, MD 20876, USA; rborrega@zalgenlabs.com (R.B.); akoval@zalgenlabs.com (A.P.K.); mheinrich@zalgenlabs.com (M.L.H.); mrowland@zalgenlabs.com (M.M.R.); rlathigra@zalgenlabs.com (R.L.); sophia.koval@gmail.com (S.A.K.); zbranco@zalgenlabs.com (Z.L.B.); 2Zalgen Labs, LCC, Broomfield, CO 80045, USA; dnelson@zalgenlabs.com (D.K.S.N.); dbush@zalgenlabs.com (D.J.B.); iaimukanova@zalgenlabs.com (I.A.); wphinney@zalgenlabs.com (W.N.P.); 3Department of Microbiology and Immunology, School of Medicine, Tulane University, New Orleans, LA 70112, USA; nbond@tulane.edu (N.G.B.); ahoffman@tulane.edu (A.R.H.); asmither@tulane.edu (A.R.S.); abell13@tulane.edu (A.R.B.-K.); lmelnik@tulane.edu (L.I.M.); kgenemar@tulane.edu (K.J.G.); kchao1@tulane.edu (K.C.); jharrel3@tulane.edu (J.E.H.); 4Bioinnovation Program, Tulane University, New Orleans, LA 70118, USA; 5Heart and Vascular Institute, John W. Deming Department of Medicine, School of Medicine, Tulane University, New Orleans, LA 70112, USA; psnarski@tulane.edu; 6Department of Physiology, School of Medicine, Tulane University, New Orleans, LA 70112, USA; 7Department of Microbiology, Tulane National Primate Research Center, Covington, LA 70433, USA; amelton@tulane.edu; 8Department of Pediatrics, School of Medicine, Tulane University, New Orleans, LA 70112, USA; areyna@tulane.edu (A.A.S.); dholton@tulane.edu (D.H.E.); jrouelle@tulane.edu (J.A.R.); jschieff@tulane.edu (J.S.S.); jrobinso@tulane.edu (J.E.R.); 9Department of Tropical Medicine, School of Public Health and Tropical Medicine, Tulane University, New Orleans, LA 70112, USA; gsabino@tulane.edu; 10Centre for Virology Research, Ribeirão Preto Medical School, University of Sao Paulo, Ribeirao Preto 14049-900, SP, Brazil; 11Department of Medicine, School of Medicine, Tulane University, New Orleans, LA 70112, USA; adrouin@tulane.edu (A.C.D.); dfusco@tulane.edu (D.N.F.); 12Eastern Polytechnic Institute, Kenema, Sierra Leone; mmomoh@tulane.edu; 13Viral Hemorrhagic Fever Program, Kenema Government Hospital, Kenema, Sierra Leone; johnatsandi@gmail.com (J.D.S.); augstgoba@yahoo.com (A.G.); robert.j.samuels@vanderbilt.edu (R.J.S.); Lansanakanneh@gmail.com (L.K.); gbakiemichael@gmail.com (M.G.); 14Ministry of Health and Sanitation, Freetown, Sierra Leone; 15Department of Biostatistics and Data Science, School of Public Health and Tropical Medicine, Tulane University, New Orleans, LA 70112, USA; jshaffer@tulane.edu; 16Department of Internal Medicine, School of Medicine, Tulane University, New Orleans, LA 70112, USA; 17Broad Institute of Massachusetts Institute of Technology (MIT) and Harvard, Cambridge, MA 02142, USA; pardis@broadinstitute.org; 18Howard Hughes Medical Institute, Chevy Chase, MD 20815, USA; 19Department of Organismic and Evolutionary Biology, Harvard University, Cambridge, MA 02138, USA; 20Department of Immunology and Infectious Diseases, Harvard T.H. Chan School of Public Health, Harvard University, Boston, MA 02115, USA; 21Department of Medicine, Division of Infectious Diseases, Massachusetts General Hospital, Boston, MA 02114, USA; 22Massachusetts Consortium on Pathogen Readiness, Boston, MA 02115, USA; 23Department of Immunology and Microbial Science, Scripps Research, La Jolla, CA 92037, USA; kga1978@gmail.com; 24Scripps Research Translational Institute, La Jolla, CA 92037, USA

**Keywords:** COVID-19 caseloads and deaths, sub-Saharan Africa, pre-existing immunity to coronaviruses, recombinant antigens, enzyme-linked immunosorbent assays, pseudovirus neutralizing antibodies, severe acute respiratory syndrome coronavirus-2, Middle Eastern respiratory syndrome coronavirus

## Abstract

Many countries in sub-Saharan Africa have experienced lower COVID-19 caseloads and fewer deaths than countries in other regions worldwide. Under-reporting of cases and a younger population could partly account for these differences, but pre-existing immunity to coronaviruses is another potential factor. Blood samples from Sierra Leonean Lassa fever and Ebola survivors and their contacts collected before the first reported COVID-19 cases were assessed using enzyme-linked immunosorbent assays for the presence of antibodies binding to proteins of coronaviruses that infect humans. Results were compared to COVID-19 subjects and healthy blood donors from the United States. Prior to the pandemic, Sierra Leoneans had more frequent exposures than Americans to coronaviruses with epitopes that cross-react with severe acute respiratory syndrome coronavirus-2 (SARS-CoV-2), SARS-CoV, and Middle Eastern respiratory syndrome coronavirus (MERS-CoV). The percentage of Sierra Leoneans with antibodies reacting to seasonal coronaviruses was also higher than for American blood donors. Serological responses to coronaviruses by Sierra Leoneans did not differ by age or sex. Approximately a quarter of Sierra Leonian pre-pandemic blood samples had neutralizing antibodies against SARS-CoV-2 pseudovirus, while about a third neutralized MERS-CoV pseudovirus. Prior exposures to coronaviruses that induce cross-protective immunity may contribute to reduced COVID-19 cases and deaths in Sierra Leone.

## 1. Introduction

A mystery surrounding the COVID-19 pandemic has been the relatively low case numbers and deaths in sub-Saharan Africa compared to other regions worldwide [1,2,3,4]. To date, the United States and several European countries (Spain, The Netherlands, Czech Republic, Belgium, Sweden, and others) have experienced more than 10,000 confirmed cases per 100,000 people. While higher COVID-19 case numbers have been reported in North Africa and Southern Africa [5], several West African nations had experienced far fewer infections, with Sierra Leone, Mali, Benin, Burkina Faso, and Nigeria all with circa 100 confirmed cases per 100,000 people or less. This trend extends to several countries in Middle Africa, including the Democratic Republic of the Congo, Chad, and Niger. Underreporting of cases due to insufficient testing is a likely factor in the lower COVID-19 numbers reported in sub-Saharan Africa. However, a major excess mortality beyond that expected for the region has not been observed between March 2020 and August 2021, arguing against large numbers of missed cases. Another also imperfect means of estimating COVID-19 caseloads is via serosurveys [3,6]. A recent study in Sierra Leone provided serological evidence that the number of SARS-CoV-2 infections is low relative to countries in Europe and the Americas [7].

Several factors, including climate and a younger population, could contribute to the reduced prevalence of COVID-19 cases and deaths in sub-Saharan Africa [8], although age-mediated protection may decline as the Delta variant circulates more widely. Four known seasonal hCoVs, including alphacoronaviruses hCoV-NL63 and hCoV-229E and betacoronaviruses hCoV-OC43 and hCoV-HKU1, cause generally mild respiratory illnesses with peak incidences during the winter months in countries with temperate climates [9]. Exposure to seasonal coronaviruses induces SARS-CoV-2 cross-reactive antibodies and T-cell responses [10,11]. While contradictory evidence exists as to whether these responses protect against COVID-19 [12,13], such studies suggest that a factor that could potentially be linked to the low prevalence of COVID-19 in sub-Saharan Africa is prior exposure to coronaviruses that induce cross-protective immunity [14].

Only a limited number of studies have quantified the prevalence of infections with seasonal coronaviruses in Africans. A study in Ghana reported that 48.3% of subjects were seropositive for three seasonal coronaviruses: hCoV-OC43, hCoV-NL63, and hCoV-229E [15]. Only 4% of pediatric pneumonia admissions were associated with seasonal hCoVs in a coastal region of Kenya [16]. While coronaviruses affecting humans in the Americas, Europe, and Asia are well described, the breadth of coronaviruses that infect Africans is largely unknown and may be different than in other parts of the world. One example of a coronavirus that is present in Africa, but not in the Americas or Europe, is Middle Eastern Respiratory Syndrome (MERS) coronavirus (MERS-CoV). MERS-CoV is a virus of African and Middle Eastern dromedary camels [17,18]. Spillovers of MERS-CoV in Africa have been documented much less frequently than in the Middle East [19].

A recent study compared SARS-CoV-2 cross-reactive antibodies in pre-pandemic blood samples from residents of Tanzania, Zambia, and the United States [14]. The prevalence of SARS-CoV-2 cross-reactive antibodies was significantly higher in samples from people living in these sub-Saharan African countries compared with samples from people living in America. Although blood samples obtained before the COVID-19 pandemic are scarce, it is essential to determine pre-pandemic anti-coronavirus seroprevalence rates in other regions of Africa and elsewhere. One setting where such studies can be conducted is the Khan Center of Excellence for Viral Hemorrhagic Fever Research located at Kenema Government Hospital (KGH) in the Eastern Province of Sierra Leone. With partners in the Viral Hemorrhagic Fever Consortium (VHFC), KGH researchers have been investigating the natural history of Lassa fever [20], determining structures of Lassa virus proteins [21,22], performing genomic analyses of Lassa virus [23], characterizing humoral and cellular immune responses to Lassa virus [24,25], and developing human monoclonal antibody therapies [26,27]. In 2013, the West African Ebola outbreak originated in a region of forested Guinea located about 150 km from KGH [28]. Cohorts of Lassa fever and Ebola survivors contribute to the study of sequelae from these illnesses [29].

To characterize serological responses to coronaviruses in Sierra Leoneans, we tested blood samples collected before the reports in late 2019 of the first COVID-19 cases in Wuhan, China. We demonstrate that, prior to the COVID-19 pandemic, the percentage of Sierra Leonians with cross-reactive antibodies to SARS-CoV-2, MERS-CoV, and seasonal coronaviruses was higher than in United States blood donors.

## 2. Materials and Methods

### 2.1. Human Subjects

All subjects enrolled in this study and/or their legal guardians provided written informed consent. Excess clinical samples (deidentified, surplus diagnostic samples) were obtained under a waiver of consent. Plasma or serum samples were obtained from Lassa fever or Ebola survivors and their contacts between September 2016 and April 2019. Only Sierra Leonean medical personnel staff were involved in the administration of health care to patients. Samples were collected from Lassa fever and Ebola survivors and their contacts in Kenema, Kailahun, and Kono Districts in the Eastern Province of Sierra Leone between September 2016 and April 2019. Blood samples from individuals with polymerase chain reaction (PCR)-confirmed COVID-19 were collected in New Orleans. Panels of serum from anonymous healthy United States blood donors were obtained from BioIVT (Westbury, NY, USA).

### 2.2. Coronavirus ELISAs

The ReSARS™ CoV-2 nucleoprotein (N) IgG ELISA developed by Zalgen Labs uses recombinant full-length HIS-tagged N of SARS CoV-2 (accession number: YP_009724390.1) produced in *Escherichia coli* (*E. coli*) BL21(DE3) pLysS cells as the immobilized viral antigen in test wells. A similar approach was used to express the N proteins of SARS-CoV (AY278741), MERS-CoV (AWH65950), hCoV-229E (DQ243941), hCoV-NL63 (DQ445912), and HCoV-OC43 (KF963200). Nucleotide sequences were codon optimized for increased expression in *E. coli*. Synthetic codon-optimized genes inserted into the pcDNA3.1IntA expression vector were used to produce antigens for the S2 and RBD ELISA. HexaPro S [30] was transiently transfected into FreeStyle 293-F cells. Four days after transfection, clarified supernatants were passed through a 0.22 μm filter and then over StrepTactin resin (Qiagen Hilden, Germany). Proteins were eluted from the column with 2.5 mM desthiobiotin and further purified by size-exclusion chromatography. RBD residues 319–591 were transiently transfected into Expi293 cells. Five days after transfection, clarified supernatants were passed through a Pall PDH4 depth filter followed by a 0.22 μm sterile filter. The clarified supernatant was concentrated using a Pall 5kDaTangential flow filtration (TFF) device and diafiltrated into 1xPBS prior to purification using Ni-NTA resin (Qiagen, Hilden, Germany). Protein was eluted from the column with 500 mM imidazole, pH 8.0. The protein was diafiltrated into 1xPBS + 10%glyerol, 0.2 µm filtered, flash frozen, and stored immediately at −80 °C. S and RBD ELISA showed >99% specificity when run on U.S. normal blood samples.

Coronavirus antigens were coated at 200 ng/well in 96-well microtiter plates (Nunc A/S, Denmark) using Carb-Bicarb buffer, pH 9.6. After antigens were immobilized, the coated microwell plates were stabilized using a proprietary blocking solution, dried, and packaged with desiccant. For the ELISA, well characterized samples with sufficient anti-N antibody titers were used as assay reference standards and similarly diluted to create reference curve dilutions. Samples were diluted 1:101 in sample diluent prior to assay. Calibrator (or reference) dilutions, diluted negative control, and samples were transferred (0.1 mL/well) in duplicate wells. Microwell plates were incubated at ambient temperature (18–30 °C) for 30 min. Microwell plates were washed four times with PBS-Tween wash buffer. Anti-Hu IgG–horseradish peroxidase conjugate reagent (Jackson ImmunoResearch, West Grove, PA, USA) was added to a microwell plate (0.1 mL/well) followed by a 30 min incubation at ambient temperature. After repeating the PBS-Tween wash, 3,3′,5,5′-tetramethylbenzidine (TMB) substrate (Moss Biotech Inc., Hanover, MD, USA) was added to each well (0.1 mL/well). The TMB substrate was incubated for 10 min followed by the addition (0.1 mL/well) of stopping solution (0.36 N sulfuric acid). Developed ELISA plates were read at 450 nm (with a 650 nm reference).

### 2.3. Pseudovirus Assays

Lentivirus-based SARS-CoV-1 and SARS-CoV-2(D614G) *Renilla* luciferase reporter virus particles (RVPs) as well as 293T-hsACE2 cells were purchased from Integral Molecular (Philadelphia, PA, USA) and used according to the manufacturer’s recommendations. Viral stocks were titrated and verified by neuralization with anti-SARS-CoV-1/2 S RBD antibody, clone VHH 72 (R&D Systems, Cambridge, MA, USA). A selected amount of virus (resulting in ~3.0–5.0 × 10^5^ RLU per point of experiment) was mixed with the experimental serum diluted 10, 20, 40, and 80 times in DMEM high glucose basal medium (Thermo Fisher, Watham, MA) in a total volume of 100 uL. After incubation for 1 h at 37 °C, each mixture was slitted equally into two wells of a 96-well plate. Then, 2.0 × 10^4^ of 293T-hsACE2 cells in 100 uL of DMEM/10% FBS was added to each well and incubated for 72 h in 6% CO_2_, 37 °C incubator. The plate was centrifuged at 2000 RPM for 5 min, the supernatant was removed, and 60 uL of Renilla-Glo substrate (diluted 1:200) was added. Luminescent reading was performed after 10 min incubation using a Spark (Tecan, Männedorf, Switzerland) plate reader.

VSV-MERS-CoV-S pseudoviral particles were generated and analyzed as published [31]. A codon-optimized S gene from MERS-CoV Florida isolate (GenBank accession number: KJ829365.1) was synthesized by Twist Bioscience (San Francisco, CA, USA) and cloned into a pcDNA3intron expression vector. The MERS S gene was expressed in 293 cells and pseudoparticles were generated using a VSVΔG*-luciferase working solution. Viral stocks were titrated and verified by neutralization with MERS Coronavirus Spike Antibody (Invitrogen, MA5-29975). A selected amount of virus (resulting in ~6.0 × 10^5^ RLU per point of experiment) was mixed with the experimental serum diluted 15, 45, 135, and 405 times in DMEM high glucose basal medium (Thermo Fisher, Waltham, MA, USA) in a total volume of 100 uL and incubated at room temperature for 30 min. Each mixture was slitted equally into three wells of a 96-well plate (25 uL/well) preplated with 1.0 × 10^4^ Vero 76 cells the day before and incubated for 1 h in 6% CO_2_, 37 °C incubator. After the addition of 25 uL/well of DMEM/10% FBS, incubation was continued overnight. Then, 50 uL of reconstituted and additionally diluted five times Bright-Glo substrate (Promega, Madison, WI, USA) was added to each well and luminescent reading was performed after 4 min incubation using the Spark plate reader.

### 2.4. Data Analysis and Statistical Methods

Laboratory data, including absorbance values, were expressed as mean ± standard error of the mean. Data were analyzed in their individual forms and were not transformed. Two-sample t tests were used to compare absorbance measures between samples from Sierra Leoneans, New Orleans COVID-19 subjects, and the controls. Ordinary linear regression models were used to compare continuous measures between lineage groups and optical density values between ELISA approaches. Pearson’s correlation coefficients or coefficients of determination were used to quantify the magnitude of linear association for linear regression approaches. Data were analyzed using Microsoft Excel (Microsoft, Redmond, WA, USA), JMP software (version 13.0.0, SAS Institute, Inc., Cary, NC, USA), and Prism (version 6.07, GraphPad Software, Inc., San Diego, CA, USA). Analyses were two-tailed with a significance threshold set at *p* < 0.05.

## 3. Results

### 3.1. Development of Recombinant Protein Coronavirus ELISA

Recombinant full-length HIS-tagged SARS CoV-2 N was produced in *E. coli* BL21(DE3) pLysS cells. SARS-CoV, MERS-CoV, hCoV-229E, hCoV-NL63, and hCov-OC43 N were produced similarly (Figure 1 and Appendix A). hCoV-HKU1 N was also cloned, but expression levels were not sufficient to include in the current study. The N proteins were purified using Ni-NTA resin (Figure 1A). Appropriately pooled fractions were pooled and analyzed by gel electrophoresis (Figure 1B–E). Codon-optimized genes for SARS-CoV-2 S2 and RBD were inserted into the pcDNA3.1IntA expression vector and used to produce antigens. Optimal conditions for coating the recombinant coronavirus antigens in 96-well microtiter plates were determined.

Cutoffs for the SARS-COV-2 N, RBD, and S2 ELISA were defined using a set of 120 samples from COVID-19 subjects from New Orleans and compared to a panel of 71 healthy United States blood donors. Receiver operator characteristic (ROC) curves were constructed to define sensitivities and specificities of the ELISAs (Appendix A). For the purpose of this analysis, a cutoff of three standard deviations above optimal sensitivity–specificity absorbance was used for the SARS-CoV-2 N, RBD, and S2 ELISAs (Appendix A). This same approach was used for the SARS-CoV N and MERS-CoV N ELISAs (Appendix A). ROC curves were also calculated for the hCoV229E, hCoV-NL63, and hCov-OC43 ELISAs. Cutoffs for these assays were set at the absorbance that optimizes sensitivity–specificity without adjustment.

### 3.2. Comparison of Serological Responses to Coronavirus Antigens between Sierra Leoneans, COVID-19 Subjects, and United States Normal Blood Donors

A set of plasma or serum samples from Sierra Leonean Lassa fever and Ebola survivors and contacts of Ebola patients (n = 120) were assessed using ELISAs that detect the binding of IgG to coronavirus N, and this binding was compared to samples from New Orleans COVID-19 subjects (n = 27) and healthy blood donors (n = 79) from the United States (Figure 2, Appendix A). Using conservative cutoffs for the recombinant antigen ELISA, we determined that 52% (62 of 120) of blood samples collected from Sierra Leoneans prior to the COVID-19 pandemic contained antibodies that were reactive to SARS-CoV-2 N (Figure 2A). Moreover, 43% (52 of 120) of the Sierra Leoneans samples contained antibodies that bound to SARS-CoV N (Figure 2B) and 54% (65 of 120) contained MERS-CoV N reactive antibodies (Figure 2C). Of the samples, 43% were from Ebola survivors, 35% were from Lassa fever survivors, and 22% were from Ebola contacts. We did not examine cross-reactivity in contacts of Lassa fever patients. There were no statistically significant differences in IgG reactivity toward antigens of any coronavirus between any of those subgroups. As expected, high percentages (>85%) of COVID-19 subjects produced antibodies reactive to N of SARS-CoV-2 (Figure 2A) and SARS-CoV (Figure 2B), but only 12% produced cross-reactive antibodies reacting with MERS-CoV N (Figure 2C). As an internal control, a sample from an immunosuppressed COVID-19 survivor, who did not produce antibodies reacting against any coronavirus protein, was included [32]. Five percent or fewer of the healthy blood donors from the United States possessed SARS-CoV-2 N, SARS-CoV N, and MERS-CoV-2 N cross-reactive antibodies (Figure 2A–C). More Sierra Leonean samples than United States samples contained antibodies that reacted with SARS-CoV-2 RBD (Figure 2B) and S2 (Appendix A) The percentages of Sierra Leoneans who showed reactivity to N of seasonal CoVs, NL63-hCoV, 229E-hCoV, and OC43-hCoV were also higher than those of the healthy United States blood donors (Figure 2D–F). A higher percentage of COVID-19 subjects reacted to N of 229E-hCoV than the other cohorts (Figure 2D). The percentages of COVID-19 subjects and healthy blood donors with antibodies reacting to NL63-hCoV and OC43-hCoV were similar.

### 3.3. Correlations of the Serological Responses of Sierra Leoneans to Coronavirus Antigens

Absorbance values with samples from Sierra Leoneans for binding to SARS-CoV-2 N were compared with binding to other coronavirus proteins. (Figure 3). As expected, there was a correlation (Pearson’s correlation = 0.71, *p* < 0.0001) between the binding of antibodies from Sierra Leoneans to SARS-CoV-2 N and SARS-CoV N (Figure 3A). This moderate correlation reflects the fact that antibodies in some samples bound more strongly to one N, but weakly or not at all to the other protein. There was also a correlation (Pearson’s correlation = 0.33, *p* = 0.0002) between binding to SARS-CoV-2 N and SARS-CoV-2 RBD (Figure 3B) and a correlation (Pearson’s correlation = 0.51, *p* < 0.0001) between binding to SARS-CoV-2 N and SARS-CoV-2 S2 (Appendix A). While most reactive Sierra Leonean samples bound both N and RBD, some samples only bound one protein or the other (Appendix A). There were also correlations between the binding of antibodies from Sierra Leoneans between SARS-CoV-2 N and N of MERS-CoV, 229E-hCoV, NL63-hCoV, and OC43-hCoV (Figure 3C–F).

### 3.4. Correlations of the Serological Responses of COVID-19 Subjects to Coronavirus Antigens

Absorbance values of samples from COVID-19 subjects for binding to SARS-CoV-2 N were compared with binding to other coronavirus proteins (Figure 4). There was a stronger correlation between reactivity to SARS-CoV-2 N and SARS-CoV N (Figure 4A) than was observed for the Sierra Leonian samples (Figure 3A). The higher Pearson’s correlations for the COVID-19 subjects vs. Sierra Leoneans (0.99 vs. 0.71) for the SARS-CoV-2 N vs. SARS-CoV N binding correlations reflects that the survivors’ antibodies recognize the precise SARS-CoV-2 N epitopes to which the subjects were exposed, and that these epitopes are highly conserved on SARS-CoV N. Correlations with reactivity to SARS-CoV-2 RBD (Figure 4B) and S2 (Appendix A) were also observed in the COVID-19 survivor samples, with Pearson’s correlations of 0.81 (*p* < 0.0001) and 0.74 (*p* < 0.0001), respectively. Most, not all, COVID-19 subjects had antibodies that reacted to SARC-CoV-2 N and S (Appendix A). This is expected as the COVID-19 subjects were simultaneously exposed to both N and S during their illness. There were also correlations between reactivity to SARS-CoV-2 N and MERS-CoV N (Figure 4C) and the alphacoronavirus hCoV-NL63 (Figure 4E). Correlations between the reactivity of antibodies to SARS-CoV-2 N and hCoV-229E N (Figure 4D) or hCoV-OC43 (Figure 4F) in samples from United States COVID-19 subjects were not observed.

### 3.5. Correlations of the Serological Responses of United States Normal Blood Donors to Coronavirus Antigens

Absorbance values of samples from United States normal blood donors for binding to SARS-CoV-2 N were compared with binding to other coronavirus proteins (Figure 5). Although the number of United States blood donors that demonstrated reactivity to SARS-CoV-2 N was low, there were correlations with reactivity to SARS-CoV N (Figure 5A), SARS-CoV-2 RBD (Figure 5B), and SARS-CoV-2 S2 (Appendix A), with Pearson’s correlations of 0.47 (*p* < 0.0001), 0.34 (*p* = 0.002), and 0.57 (*p* < 0.0001), respectively. A minority of the reactive samples from United States blood donors reacted to both SARS-CoV-2 N and RBD (Appendix A). Correlations between the reactivity of antibodies to SARS-CoV-2 N and MERS-CoV N (Figure 5C), hCov-NL63 (Figure 5E), or hCoV-OC43 (Figure 5F) in samples from United States blood donors were not observed. However, a correlation was observed between the reactivity of antibodies to SARS-CoV-2 N and alphacoronavirus hCov-229E N (Figure 5D) in samples from United States blood donors (Pearson’s correlation = 0.47, *p* < 0.0001). Some of the samples from the normal blood donors that reacted with SARS-CoV-2 N were among those with the highest reactivity to hCoV-229E (Figure 5D, blue squares).

### 3.6. Neutralization of Pseudoviruses Expressing Spike of SARS-CoV-2 or MERS-CoV by Antibodies in Pre-Pandemic Blood Samples of Sierra Leoneans

Pseudoviruses expressing S of SARS-CoV-2 or MERS-CoV-2 were employed to assess the ability of antibodies from Sierra Leoneans to neutralize these betacoronaviruses (Figure 6). Only samples with sufficient volume were tested by neutralization. Most samples, including those that bound RBD or S2, failed to neutralize either virus. However, a subset of samples tested did neutralize them (Appendix A): 12 of 47 (26%) samples tested neutralized SARS-CoV-2 pseudoviruses, while 10 of 30 samples tested neutralized MERS-CoV pseudoviruses. Most samples only neutralized at a single dilution. Two samples (SL6 and SL25) neutralized SARS-CoV-2 pseudovirus and MERS-CoV pseudoviruses at 1:10 and 1:15 dilutions, respectively. An examination of the samples that neutralized SARS-CoV-2 or MERS-CoV-2 pseudoviruses at more than a single dilution showed that neutralization titers were lower against SARS-CoV-2 pseudoviruses (Figure 6A–C) than against MERS-CoV-2 pseudoviruses (Figure 6D–F). The failure of blood samples to cross-neutralize 100% of viruses even at high antibody levels has been noted in studies of other viruses [33,34].

### 3.7. Demographics of the Coronavirus Serological Responses in Sierra Leoneans

We assessed the age distribution of subjects with positive serological responses to SARS-CoV-2 N and N of other coronaviruses (Appendix A, Appendix A). No significant trends for coronavirus serology were observed across of the age groups for any of the coronaviruses tested, which suggests that exposures in Sierra Leoneans occur throughout the lifespan. Serological responses to SARS-CoV-2 N and N of other coronaviruses also did not differ by sex (Appendix A). The distribution of positive samples for seroreactivity to SARS-CoV-2 N and N of other coronaviruses over the time of collection also appeared to be random (Appendix A). Seroreactivity may be short-lived with seasonal and other coronaviruses, including SARS-CoV-2 [35,36]. We examined the duration of antibody seroreactivity in a limited subset of samples over two-year periods (Appendix A). Reactivities either increased or were stable for most coronaviruses in these subjects.

## 4. Discussion

We examined serological responses to coronaviruses in blood samples from Sierra Leoneans that were collected before the reports in late 2019 of the first COVID-19 cases in Wuhan, China. Prior to the pandemic, Sierra Leoneans appear to have had more frequent exposures than Americans to coronaviruses with SARS-CoV-2, SARS-CoV, and MERS-CoV cross-reactive epitopes. An alternative explanation for our results is that the immunological repertoires of Sierra Leoneans produce more broadly cross-reactive antibodies upon coronavirus infection and/or that the duration of these immune responses is longer than in Americans. It is unlikely that the reactivities to SARS-CoV-2, SARS-CoV, and MERS-CoV N proteins measured by our recombinant antigen ELISAs are spurious. Conservative cutoffs were applied to the data analysis, and the assays displayed expected sensitivities and specificities on samples from COVID-19 subjects. We also observed a correlation between reactivity to N and RBD or S2 subunit of S with Sierra Leonean samples. Similar correlations between seroreactivities to SARS-CoV-2 N and RBD were observed in COVID-19 subjects and United States normal blood donors, although higher and lower percentages of the samples were positive in the respective cohorts. Correlation for reactivities to these immunologically distinct antigens would not be expected to occur by chance and likely resulted from prior coronavirus infections. A lower percentage of United States blood donors reacted to N proteins of seasonal coronaviruses, suggesting that a higher portion of Sierra Leoneans have exposure to coronaviruses compared to Americans.

Correlations were also observed between seroreactivities to SARS-CoV-2 N and N of other coronaviruses in the cohorts examined. The relationship of the correlations to sequence conservation amongst coronavirus proteins is presently unclear. The amino acid sequence similarity of N is conserved across the *Coronaviridae* family (Figure 7). It can be seen in Figure 7A that 97% of the SARS-Cov-2 N amino acids are chemically similar to those of SARS-CoV N and 91% are identical. A high degree of similarity can also be observed between SARS-CoV-2 N and MERS CoV N. The N proteins of the alphacoronaviruses 229E-hCoV and NL63-hCoV share a similar level of identical and chemical similarity in amino acids to SARS-CoV-2, SARS-CoV, and MERS-CoV N as the betacoronaviruses OC43-hCoV and HKU-1-hCoV. The amino-terminus of the N proteins, which contains the RNA-binding domain [37], is somewhat more conserved than the carboxyl-terminal dimerization domain [38] (Figure 7B,C). Therefore, exposures to either alpha- or betacoronaviruses could contribute to the generation of SARS-CoV-2, SARS-CoV, and MERS-CoV N cross-reactive antibodies. There is also sequence conservation between S of known alphacoronaviruses and betacoronaviruses (Figure 7D), but less than for the N. This similarity varies across S. The S1 subunit, which binds receptors, is more divergent than the highly conserved S2 subunit, which is involved in fusion (Figure 7E,F). Angiotensin-converting enzyme 2 (ACE2) is the cellular receptor used by SARS-CoV-2 and SARS-CoV. Although the alphacoronavirus hCoV-NL63 uses ACE2 as its receptor and binds to common ACE2 regions [39,40], there is no structural similarity between alphacoronavirus and betacoronavirus RBDs. Reactivity of antibodies produced by Sierra Leonians to the RBD and S2 of SARS-CoV2 is likely due to exposure to betacoronaviruses S that share common epitopes.

Prior studies have demonstrated the potential for infections with seasonal coronaviruses to induce cross-reactive immune responses to SARS-CoV-2 proteins. Antibodies to seasonal hCoVs and the betacoronaviruses MERS-CoV and SARS-CoV can cross-react with various SARS-CoV-2 antigens [41,42,43]. Anderson et al. [13] found that 4% of serum samples collected in 2017 contained antibodies that reacted to SARS-CoV-2 S, 1% reacted to RBD, and 16% reacted to N. Titers of these antibodies were boosted upon SARS-CoV-2 infection. The lower percentage of US normal blood donors (5%) found here that bound SARS-Cov-2 may be due to the conservative approach used to determine assay cut-offs. Another prior study by Ng and co-workers [44] found that some United Kingdom citizens previously exposed to seasonal coronaviruses have antibodies targeted to the conserved S2 subunit of S. Pre-existing memory B cells that are cross-reactive with other coronaviruses are activated on SARS-CoV-2 infection [45]. Monoclonal antibodies (mAbs) that cross-react with SARS-CoV-2 proteins were isolated from these memory B cells.

Among Sierra Leonian blood samples tested in neutralization assays, approximately a quarter possessed a low titer of neutralizing antibodies against SARS-CoV-2 pseudovirus, while approximately a third had low titers of MERS-CoV neutralizing antibodies. Prior studies have generally found a lack of SARS-CoV-2 cross-neutralization activity in pre-pandemic sera from patients with prior PCR-confirmed seasonal coronavirus infection [46]. Likewise, Anderson et al. [13] found very low or undetectable levels of SARS-CoV-2 neutralizing antibodies in pre-pandemic blood samples regardless of whether the sample possessed cross-reactive antibodies against SARS-CoV-2 S or N proteins. In contrast, Ng et al. [44] found that most sera from SARS-CoV-2–uninfected children or adolescents with cross-reactive antibodies also neutralized SARS-CoV-2, albeit on average less potently than COVID-19 patient sera. In contrast to the results of cross-reactivity studies in Sierra Leoneans, most of the cross-reactivity detected by Ng and co-workers in subjects from the United Kingdom was directed to the more highly conserved S2 subunit of S [44]. Further studies on Sierra Leonean pre-pandemic samples that neutralize SARS-CoV-2 and MERS-CoV pseudoviruses are required to determine the epitope specificities of these antibodies.

Conflicting results have been obtained regarding whether pre-existing humoral immunity induced by infection with seasonal coronaviruses confers protection against SARS-CoV-2 infection or COVID-19 severity. Whereas some studies have found that pre-existing immunity against endemic human coronaviruses was not associated with protection against SARS-CoV-2 infections or hospitalizations [13,42], another study found mitigation of disease manifestations [12]. On the other extreme, the presence of antibodies against NL63 and 229E N was correlated with increased clinical severity in another study [47]. Potentially, prior immunity to a SARS-like CoV in the Sierra Leonean population may mitigate against SARS-CoV-2 spread or the induction of severe disease. A similar hypothesis was proposed by Tso et al. [14] upon analysis of pre-pandemic blood samples from Tanzania and Kenya. Further studies will be required to determine whether the immunological responses detected here are correlated with protection from SARS-CoV-2 infection or disease. Additional challenges exist in conducting serosurveys in Africa [6,48,49]. This potential pre-existing immunity is currently not well enough understood to negate efforts at vaccine delivery, and as mentioned above, may be less protective against the Delta variant than initial SARS-CoV-2 variants. While this question is important to examine, at present the low prevalence of COVID-19 in Sierra Leone precludes epidemiological studies to directly address this question.

Elucidating the diversity of coronaviruses that infect Sierra Leoneans and their natural history in this population is a priority for assessing whether preexisting immunity may have provided population-level protection against COVID-19. While there has been limited sampling of coronaviruses in African wildlife or other animals, it is apparent that there is a large as yet unsampled diversity. Coronaviruses related to SARS-CoV-2, SARS-CoV, seasonal coronaviruses NL63 and 229E, and MERS-CoV have been detected by genomic sequencing in bats in Ghana [50,51] proposed to be ancestors of these hCoVs from previous spillover events [52]. For example, hCoV229E may have originated from the large diversity of coronaviruses in hipposiderid bats (genus *Hipposideros*) with camelids as possible intermediate hosts [53].

Examination of whether exposure to circulating coronaviruses provides African populations immunological protection against diseases caused by other coronaviruses should also consider the possibility of a protective effect against MERS. MERS-CoV is a virus of dromedary camels [17]. More than 85% of the world’s dromedary camels are found in Africa, principally the East African countries of Sudan, Somalia, Ethiopia, and Kenya [54]. However, symptomatic cases of MERS in Africans are rare. To date, all cases of MERS have occurred in the Middle East or as a result of importation by travelers from the Middle East [19,55,56]. In many Middle Eastern countries, MERS-CoV antibodies have been found in most African camels, which may suggest that they have been infected with MERS-CoV or a related coronavirus [57,58]. Despite the high seroprevalence among African camels, seroprevalence in Africans with occupational exposure to camels is low [54,59]. In contrast, in Middle Eastern countries where seroprevalence has been measured, MERS-CoV antibodies were increased in camel shepherds and slaughterhouse workers [55,60,61,62]. Strain differences in the MERS-CoV circulating in African camels could be a factor in the low incidence of MERS in Africans [63]. Further studies are required to determine the basis for the low incidence of MERS in Africans living in counties where exposure to camels is common.

There are a number of limitations to the current study. The limited availability of samples with sufficient volume precluded more extensive characterizations of the coronavirus-reactive antibodies, including the determination of end point titers. Another limitation was that the duration of coronavirus-specific immune responses and the epitope specificity of the antibodies were not captured and were beyond the scope of the study. The samples examined here were collected only from the Eastern province of Sierra Leone. Similar analyses with pre-pandemic samples should be performed with samples from other regions of Sierra Leone and from other countries in West Africa. Other limiting factors are the lack of information on the frequencies of respiratory illnesses in Sierra Leoneans across the lifespan, and knowledge regarding the types of coronaviruses and other respiratory pathogens that infect Sierra Leoneans.

## 5. Conclusions

Pre-existing immunity to coronavirus antigens should be further investigated as a potential factor contributing to reduced caseloads and deaths from COVID-19 in Sierra Leone. It is likely that humans in Sierra Leone are frequently exposed to SARS-related and MERS-related viruses. Studies should be conducted to fully characterize immune responses directed against coronaviruses by Sierra Leoneans. Several studies have reported T cell reactivity against SARS-CoV-2 in people with no known exposure to the virus, which may in part be related to prior exposure to seasonal coronaviruses. It is possible that cellular immunity to endemic coronaviruses also has a protective role. Studies to access the prevalence of humoral and cellular immunity to coronaviruses are needed in other African countries with low reported incidence of SARS-CoV-2 infections. Equitable COVID-19 vaccine distribution should continue even in countries with currently low numbers of cases and deaths. The role of both natural immunity and vaccine-induced immunity should be investigated in these populations. Another priority for future research will be to define the diversity of coronaviruses that circulate in humans or frequently spillover from animals to humans living in Sierra Leone and other West African countries.

## Figures and Tables

**Figure 1 viruses-13-02325-f001:**
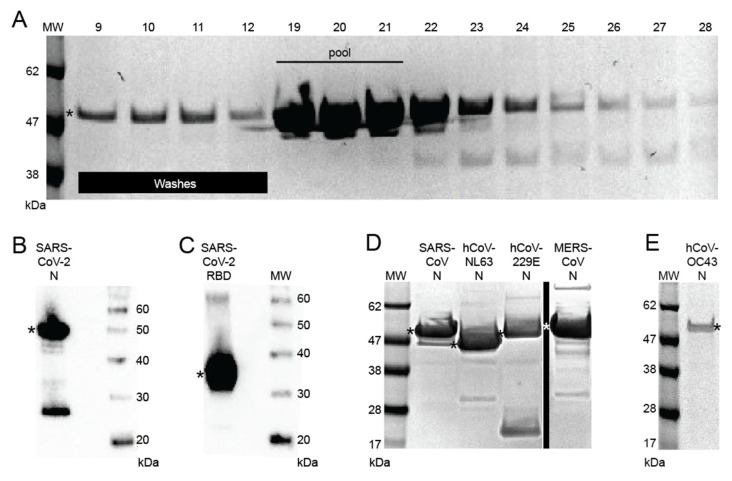
Recombinant coronavirus proteins. (**A**) Nucleoprotein (N) from SARS-CoV expressed in *E. coli*, purified, and proteins in selected fractions were resolved by sodium dodecyl-sulfate polyacrylamide gel electrophoresis (SDS-PAGE). (**B**) SARS-CoV-2 nucleoprotein was expressed in *E. coli*, purified, and resolved by SDS-PAGE as described for SARS-CoV. Monomeric forms of the protein (asterisks) and a breakdown product were detected. (**C**) SARS-CoV-2 receptor binding domain was expressed in Expi293 cells, purified, and resolved by SDS-PAGE for (**D**) SARS-CoV N, hCoV-NL63 N, hCoV-229E N, MERS-CoV, and (**E**) hCoV-OC43 N. Vertical lines indicate removal of lanes from a gel. Asterisk (*): monomeric N or RBD. Uncropped polyacrylamide gel D is shown in Appendix A.

**Figure 2 viruses-13-02325-f002:**
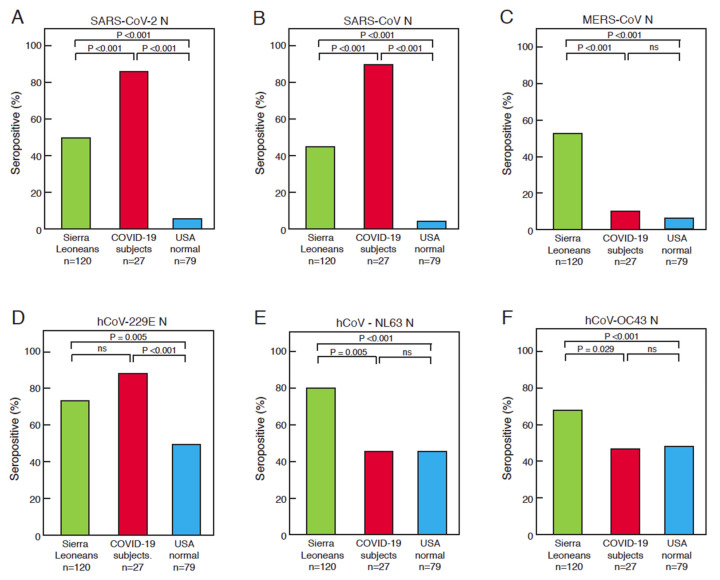
Serological responses to coronavirus antigens by Sierra Leoneans, COVID-19 subjects, and United States normal blood donors. The percentage of positive results for the indicated coronavirus proteins for the three cohorts are shown: (**A**) SARS-CoV-2 N, (**B**) SARS-CoV-2 RBD, (**C**) MERS-CoV-N, (**D**) hCoV-229E N, (**E**) hCoV-NL63 N, and (**F**) hCoV-OC43 N.

**Figure 3 viruses-13-02325-f003:**
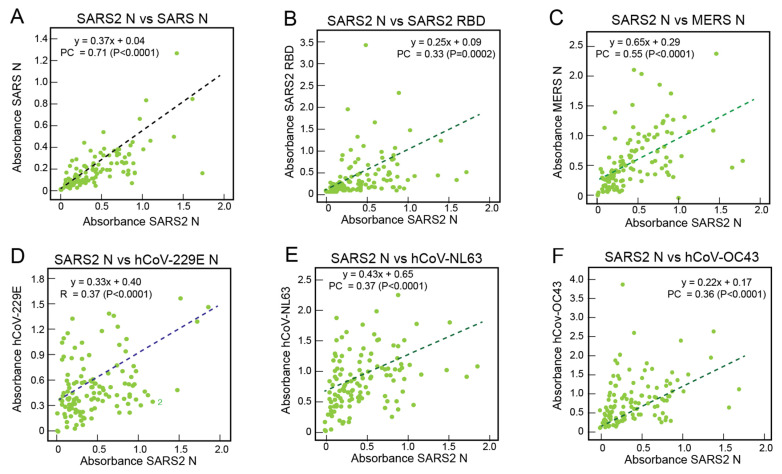
Correlations of the serological responses of Sierra Leoneans to coronavirus antigens. Binding of IgG in serum or plasma samples (1:100 dilution) from Sierra Leoneans was quantified using ELISA coated with recombinant coronavirus proteins. Reactivity to SARS-CoV-2 N was compared to other coronavirus proteins in each panel: (**A**) SARS-CoV N, (**B**) SARS-CoV-2 RBD, (**C**) MERS-CoV-N, (**D**) hCoV-229E N, (**E**) hCoV-NL63 N, and (**F**) hCoV-OC43 N. Dotted lines are linear regression plots of seroreactivity of SARS-CoV-2 N versus other recombinant coronavirus proteins. PC = Pearson’s correlation.

**Figure 4 viruses-13-02325-f004:**
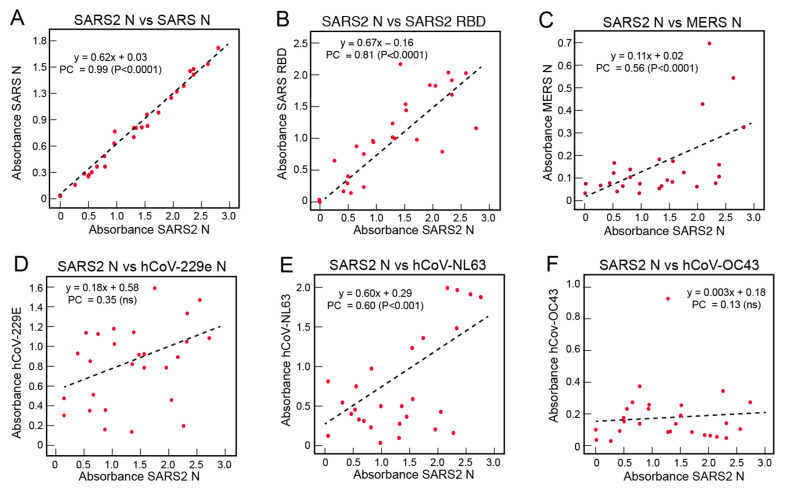
Correlations of the serological responses of COVID-19 subjects to coronavirus antigens. Binding of IgG in serum or plasma samples (1:100 dilution) from COVID-19 subjects was quantified using ELISA coated with recombinant coronavirus proteins. Reactivity to SARS-CoV-2 N was compared to other coronavirus proteins in each panel: (**A**) SARS-CoV N, (**B**) SARS-CoV-2 RBD, (**C**) MERS-CoV-N, (**D**) hCoV-229E N, (**E**) hCoV-NL63 N, and (**F**) hCoV-OC43 N. Dotted lines are linear regression plots of seroreactivity of SARS-CoV-2 N versus other recombinant coronavirus proteins. PC = Pearson’s correlation.

**Figure 5 viruses-13-02325-f005:**
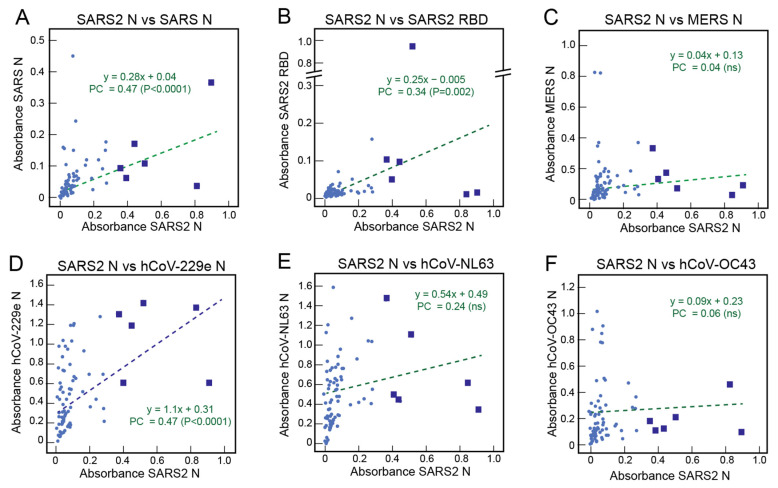
Correlations of the serological responses of United States normal blood donors to coronavirus antigens. Binding of IgG in serum or plasma samples (1:100 dilution) from United States normal blood donors was quantified using ELISA coated with recombinant coronavirus proteins. Reactivity to SARS-CoV-2 N was compared to other coronavirus proteins in each panel: (**A**) SARS-CoV N, (**B**) SARS-CoV-2 RBD, (**C**) MERS-CoV-N, (**D**) hCoV-229E N, (**E**) hCoV-NL63 N, and (**F**) hCoV-OC43 N. Dotted lines are linear regression plots of seroreactivity of SARS-CoV-2 N versus other recombinant coronavirus proteins. PC = Pearson’s correlation.

**Figure 6 viruses-13-02325-f006:**
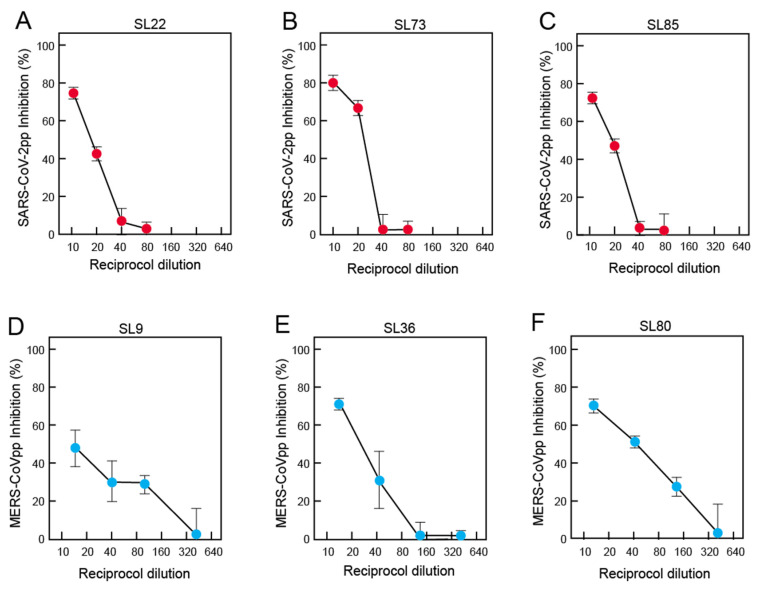
Neutralization of pseudoviruses expressing SARS-CoV-2 and MERS-CoV spike. SARS-CoV-2 pseudovirus neutralization curves were determined with samples from the following subjects: (**A**) SL22, (**B**) SL 73, and (**C**) SL83. MERS-CoV pseudovirus neutralization curves were determined with samples from the following subjects: (**D**) SL9, (**E**) SL36, and (**F**) SL 80.

**Figure 7 viruses-13-02325-f007:**
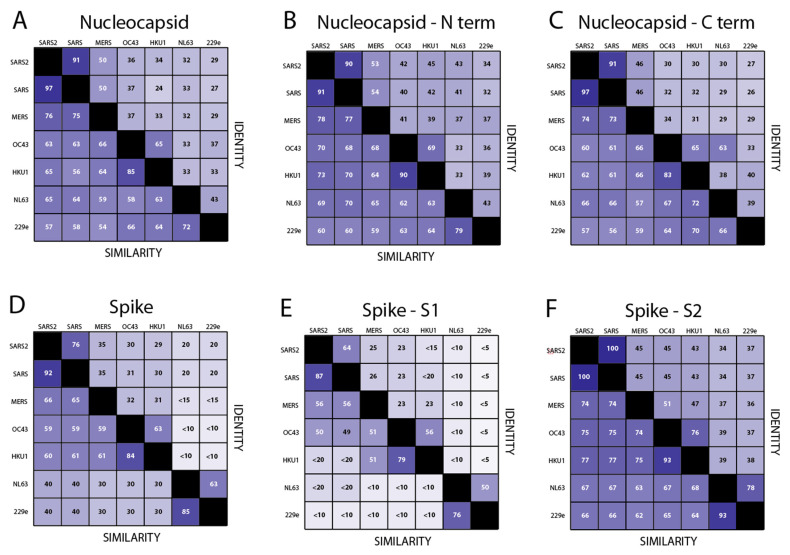
Sequence identity and similarity of nucleocapsid and spike across the *Coronaviridae* family. The percentage of identical and chemically similar amino acids are indicated in (**A**) across the entire N proteins. Shading is proportional to the percent identity or similarity. (**B**) shows the amino terminal (N-term) sequences of the N proteins. (**C**) shows the carboxyl-terminal (C-term) sequences of the N proteins (**D)** across the entire spike proteins. (**E**) S1 subunit sequence of the S proteins and (**F**) S2 subunit sequence of the S proteins.

## Data Availability

The data presented in this study is available in Appendix A.

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
