# Peer review of "Cross-Reactive Antibodies to SARS-CoV-2 and MERS-CoV in Pre-COVID-19 Blood Samples from Sierra Leoneans"

_viruses, 2021, doi:10.3390/v13112325_

Round 1

Reviewer 1 Report

The manuscript titled “Cross-reactive antibodies to SARS-CoV-2 and MERS-CoV in pre-COVID-19 blood samples from Sierra Leonians” by Borrega et al. reported pre-pandemic sera from Sierra Leoneans have relatively higher cross-reactivity against SARS-CoV-2, SARS-CoV, MERS-CoV, hCoV-229E, hCoV-NL63, and hCoV-OC43 comparing to normal sera from U.S. subjects. These sera cross-react with nucleoprotein (N) and spike protein (both receptor binding domain [RBD] and S2 region) of these coronaviruses albeit some are very weak in reactivity. Some of these sera also showed weak neutralization against SARS-CoV-2 and MERS-CoV pseudoviruses. While the authors reported interesting observation on the cross-reactivity from pre-pandemic sera, it is unclear how the cross-reactivity been established. Several concerns should be addressed before further consideration.

  1. Antibody binding replies on recognition of specific amino acid sequences and local structures. It is not sure how the cross-reactivity correlates with the sequence conservation among these coronaviruses. Although S2 region of spike proteins is more conserved across coronaviruses, it seems less cross-reactive than RBD.
  2. S2 is blurry and hardly readable. And the method described only protein production used for ELISA but not how the ELISA was performed. What secondary antibody was used for detection should be clearly addressed. Moreover, background/unspecific binding/absorbance in the ELISA seems not been subtract before data analysis.
  3. I am not sure why those datapoints showing extremely weak binding in Fig. 3/4/5 were used for data analysis. Especially, some datapoints with both x and y absorbance values approaching to zero are considered positive and included in the analysis. These data are likely not beyond the “conservative cutoffs” the authors claimed in the main text and should be excluded from data analysis in these figures if they are under the error margin.
  4. The authors have compared many correlations of cross-reactivity with different viral proteins. However, how these correlations relate to the sequence conservation have not been well analyzed. I am not sure what these data can tell us.
  5. All the neutralization curves should be plotted if there are more serum samples can neutralize SARS-CoV-2 or MERS-CoV. Given these samples are cross-reactive to many coronaviruses, it is unclear why there are no serum can neutralize both SARS-CoV-2 or MERS-CoV.
  6. There are many serum samples cross-react with SARS-CoV-2 RBD, which is much less conserved across coronaviruses. Did this mean there is SARS-CoV-2 like virus circulating in Sierra Leonians prior to the COVID-19 pandemic?
  7. Both N and spike proteins consist of multiple domains with different sequence conservation across coronaviruses. The sequence conservation comparison in Figure 7 could be more informative if comparisons were based on protein domains, which likely impact the antibody cross-reactivity.

Author Response

The manuscript titled “Cross-reactive antibodies to SARS-CoV-2 and MERS-CoV in pre-COVID-19 blood samples from Sierra Leonians” by Borrega et al. reported pre-pandemic sera from Sierra Leoneans have relatively higher cross-reactivity against SARS-CoV-2, SARS-CoV, MERS-CoV, hCoV-229E, hCoV-NL63, and hCoV-OC43 comparing to normal sera from U.S. subjects. These sera cross-react with nucleoprotein (N) and spike protein (both receptor binding domain [RBD] and S2 region) of these coronaviruses albeit some are very weak in reactivity. Some of these sera also showed weak neutralization against SARS-CoV-2 and MERS-CoV pseudoviruses. While the authors reported interesting observation on the cross-reactivity from pre-pandemic sera, it is unclear how the cross-reactivity been established. Several concerns should be addressed before further consideration.

We thank the reviewer for the expert comments and support.

  1. Antibody binding replies on recognition of specific amino acid sequences and local structures. It is not sure how the cross-reactivity correlates with the sequence conservation among these coronaviruses. Although S2 region of spike proteins is more conserved across coronaviruses, it seems less cross-reactive than RBD.

We agree that the observation that S2 seems to be less cross-reactive  is important and discuss this point in the revised manuscript.

  1. S2 is blurry and hardly readable. And the method described only protein production used for ELISA but not how the ELISA was performed. What secondary antibody was used for detection should be clearly addressed. Moreover, background/unspecific binding/absorbance in the ELISA seems not been subtract before data analysis.

We apologize for the low resolution of Fig. S2 which shows the receiver operator characteristic (ROC) curves. We have replaced it with the high-res version. Both reviewers requested additional details regarding the ELISA, which are now provided in revision. The secondary antibody used was Anti-Hu IgG-horseradish peroxidase conjugate reagent (Jackson ImmunoResearch, West Grove, PA). The background for all assays was low and subtracted prior to setting cuttoffs using ROC curves.

  1. I am not sure why those datapoints showing extremely weak binding in Fig. 3/4/5 were used for data analysis. Especially, some datapoints with both x and y absorbance values approaching to zero are considered positive and included in the analysis. These data are likely not beyond the “conservative cutoffs” the authors claimed in the main text and should be excluded from data analysis in these figures if they are under the error margin.

Datapoints approaching zero were not considered as positive. This is described in Figure 2. For the correlations in Figs 3-5 we had to include both positive and negative samples in the correlation analysis. Some samples were positive on some assays, but negative on the others. Furthermore, while some samples were positive on both assays the level of reactivity is important to consider.

  1. The authors have compared many correlations of cross-reactivity with different viral proteins. However, how these correlations relate to the sequence conservation have not been well analyzed. I am not sure what these data can tell us.

We agree that the relationship of the correlations to sequence conservation is presently unclear. We added this point in revision.

  1. All the neutralization curves should be plotted if there are more serum samples can neutralize SARS-CoV-2 or MERS-CoV. Given these samples are cross-reactive to many coronaviruses, it is unclear why there are no serum can neutralize both SARS-CoV-2 or MERS-CoV.

We plotted all of the serum samples that showed neutralization at two or more dilutions. There were two samples that neutralized both. Two samples neutralized both SARS-CoV-2 pseudoviruses and MERS-CoV at a 1:10 and a 1:15 dilution, respectively.

  1. There are many serum samples cross-react with SARS-CoV-2 RBD, which is much less conserved across coronaviruses. Did this mean there is SARS-CoV-2 like virus circulating in Sierra Leonians prior to the COVID-19 pandemic?

We believe that it is likely that SARS-related and MERS-related viruses circulate in humans in Sierra Leone. This point is made more explicit in revision.

  1. Both N and spike proteins consist of multiple domains with different sequence conservation across coronaviruses. The sequence conservation comparison in Figure 7 could be more informative if comparisons were based on protein domains, which likely impact the antibody cross-reactivity.

We revised figure 7 to show both the sequence conservation overall as well as in the major domains of nucleoprotein and spike.

Reviewer 2 Report

The manuscript "Cross-reactive antibodies to SARS-CoV-2 and MERS-CoV in pre-COVID-19 blood samples from Sierra Leonians" by Borrega et al. (Manuscript ID: viruses-1422563) presents data implying on potential impact of preexisting immunity to the outcome of SARS-CoV-2 infection. Authors were focused on preexisting immunity raised due to exposure to coronaviruses other than SARS-CoV-2. As impact of preexisting immunity on susceptibility to SARS-CoV-2 infection is still far from being fully elucidated, I find presented data very important and valuable. However, there are some issues that have to be clarified/corrected (see below).

  1. Section 2.1 – The total number of sera used in the study as well as its specific parts (validation of the assay, duration of preexisting immunity, comparisons, etc.) has to be clearly indicated. In addition, authors did not specify the provider of sera of US healthy donor (… XXXX; line 152)
  2. Section 2.2 – Details on the ELISAs procedures have to be provided. IgG specific for N of SARS-CoV-2 was measured by commercial ELISA (Zalgen Labs) and IgG reactivity toward other proteins of coronaviruses was evaluated by inhouse designed ELISAs?
  3. Cut-off values for SARS-CoV-2 antigens ELISA (N, S2 and RBD) were set by using 120 sera of COVID-19 patients (line 235) and reactivity toward N proteins of -non-SARS-CoV-2 coronaviruses was assessed for 27 samples (Figure 2). Why some samples were excluded and what was the exclusion criterium?
  4. What was the selection criterium for samples analyzed by neutralization assays (SARS-CoV-2: 47/120; MERS-CoV: 30/120) as well as those used for assessment of neutralization titer (3 samples per assay)?
  5. Samples form Sierra Leone were collected from lassa fever and ebola survivors. How many sera was in each subgroup? Did authors confirm that there is no statistical significance in IgG reactivity toward specific antigens of coronaviruses between those subgroups?
  6. What was the selection criterium for subjects involved in the follow up of coronaviruses-specific seroreactivity during two-years period? Are there any data on coronavirus infections with those subjects during the follow up period?
  7. Are all sera in the study were evaluated simultaneously for reactivity toward one specific antigen? If not:

- did authors used some internal controls (positive and negative) in order to normalize the results of specific measurements? or

- did authors check interassay reproducibility of cut-off values they set?  

Namely, the levels of specific IgG are expressed as OD values. The OD values for a single sample could significantly vary for individual measurements with same kit (due to ambiental factors, duration of enzymatic reaction, etc.). The authors have to explain how they ensure valid comparability of individual measurements. This could be especially important for accurate declaring of “border” cases (samples which are not highly positive or clearly negative) as seropositive or seronegative.

  1. I find Pearson´s correlation analysis to be more convenient than linear regression analysis for estimation of mutual correlations (and their statistical significance) among analyzed IgG pools. The results of Pearson´s correlation analysis have to be provided.

Minor remarks:

  1. Except its first mentioning (line 156), Escherichia coli has to be consistently assigned trough the Manuscript as coli (in italic)
  2. Title of Section 2.2 has to be in plural
  3. Line 240: check the sentence “This same approach…”
  4. Line 254 - 255: numbers of sera from USA (COVID-19 and healthy blood donors) have to be added
  5. Line 273: “NL63 and” has to be deleted
  6. Line 311: the sentence
  7. Line 414: “found that” has to be deleted (duplicate)
  8. Line 443: “nucleocapsid” has to be deleted (abbreviation is already introduced)
  9. Line 458: check the sentence “Coronaviruses related…”

Author Response

The manuscript "Cross-reactive antibodies to SARS-CoV-2 and MERS-CoV in pre-COVID-19 blood samples from Sierra Leonians" by Borrega et al. (Manuscript ID: viruses-1422563) presents data implying on potential impact of preexisting immunity to the outcome of SARS-CoV-2 infection. Authors were focused on preexisting immunity raised due to exposure to coronaviruses other than SARS-CoV-2. As impact of preexisting immunity on susceptibility to SARS-CoV-2 infection is still far from being fully elucidated, I find presented data very important and valuable. However, there are some issues that have to be clarified/corrected (see below).

We thank the reviewer for the expert comments and support.

  1. Section 2.1 – The total number of sera used in the study as well as its specific parts (validation of the assay, duration of preexisting immunity, comparisons, etc.) has to be clearly indicated. In addition, authors did not specify the provider of sera of US healthy donor (… XXXX; line 152)

We regret the omission. Serum panels were purchased from BioIVT.

  1. Section 2.2 – Details on the ELISAs procedures have to be provided. IgG specific for N of SARS-CoV-2 was measured by commercial ELISA (Zalgen Labs) and IgG reactivity toward other proteins of coronaviruses was evaluated by inhouse designed ELISAs?

Both reviewers requested additional details regarding the ELISA, which are now provided in revision.

  1. Cut-off values for SARS-CoV-2 antigens ELISA (N, S2 and RBD) were set by using 120 sera of COVID-19 patients (line 235) and reactivity toward N proteins of -non-SARS-CoV-2 coronaviruses was assessed for 27 samples (Figure 2). Why some samples were excluded and what was the exclusion criterium?

We uses a larger number of samples to establish the cutoffs to achieve greater precision.  There were not exclusion criteria for the comparisons to the Sierra Leonean samples. These were consecutive patients admitted to the Tulane Hospital early in the pandemic. As discussed below a lower number of samples were used the comparisons with the Sierra Leoneans samples and United States blood donors because the COVID-19 samples were dispersed to different ELISA plates to control for potential interplate variability, which was not observed.

  1. What was the selection criterium for samples analyzed by neutralization assays (SARS-CoV-2: 47/120; MERS-CoV: 30/120) as well as those used for assessment of neutralization titer (3 samples per assay)?

Only samples with sufficient volume were tested by neutralization. This is indicated in revision.

  1. Samples form Sierra Leone were collected from lassa fever and ebola survivors. How many sera was in each subgroup? Did authors confirm that there is no statistical significance in IgG reactivity toward specific antigens of coronaviruses between those subgroups?

We thank the reviewer for pointing out this omission. 43% of samples were from Ebola survivors , 35% were from Lassa fever survivors and 22% were from Ebola contacts. We did not do examine ross-reactivity and Lassa fever survivors contacts. There were no statistically significance differences in IgG reactivity toward antigens of any coronavirus between any of those subgroups. This in indicated in revision.

  1. What was the selection criterium for subjects involved in the follow up of coronaviruses-specific seroreactivity during two-years period? Are there any data on coronavirus infections with those subjects during the follow up period?

The ability to do follow-up testing depended on the amount of serum available, which was limiting for most subjects and did not allow multiple ELISA or pseudovirus neutralization assays to be performed. Data on coronavirus infections is not collected in Sierra Leone. We have added each of the points in revision.

  1. Are all sera in the study were evaluated simultaneously for reactivity toward one specific antigen? If not:

- did authors used some internal controls (positive and negative) in order to normalize the results of specific measurements?

As discussed in revision each plate was run with internal controls.

or

- did authors check interassay reproducibility of cut-off values they set?  Namely, the levels of specific IgG are expressed as OD values. The OD values for a single sample could significantly vary for individual measurements with same kit (due to ambiental factors, duration of enzymatic reaction, etc.). The authors have to explain how they ensure valid comparability of individual measurements.

We agree that this is a significant point. There was no significant intra assay variability,  which was determined by running a standard curve with each assay plate. This is discussed in revision. Very similar results for the Pearson’s correlations are obtained whether OD values or units per ml values are calculated from the standard curves. We showed the OD values to allow the reader to more easily compare reactivities with the several different ELISA, which could be somewhat obscured if converting to units per ml. As mentioned above we also interspersed the samples from the each of the cohorts (Sierra Leoneans, COVID patients and US blood donors) on each plate rather than running all the samples from one cohort on the same plate.

This could be especially important for accurate declaring of “border” cases (samples which are not highly positive or clearly negative) as seropositive or seronegative.

We agree that setting the cutoff is important and would affect the overall numbers of positives samples. This is why we set a high cutoff values for the SARS and MERS assays. Had we set the cutoffs lower there were have been more positive samples that would have been borderline. This can be seen in Fig. S7. We also  included all the raw OD values for each sample in supplementary Tables S1 to S3.  Each positive sample in these tables is highlighted.

  1. I find Pearson´s correlation analysis to be more convenient than linear regression analysis for estimation of mutual correlations (and their statistical significance) among analyzed IgG pools. The results of Pearson´s correlation analysis have to be provided.

We changed the estimates of statistical significance to Pearson’s correlation analysis.

Minor remarks:

  1. Except its first mentioning (line 156), Escherichia coli has to be consistently assigned trough the Manuscript as coli (in italic)
  2. Title of Section 2.2 has to be in plural
  3. Line 240: check the sentence “This same approach…”
  4. Line 254 - 255: numbers of sera from USA (COVID-19 and healthy blood donors) have to be added
  5. Line 273: “NL63 and” has to be deleted
  6. Line 311: the sentence
  7. Line 414: “found that” has to be deleted (duplicate)
  8. Line 443: “nucleocapsid” has to be deleted (abbreviation is already introduced)
  9. Line 458: check the sentence “Coronaviruses related…”

Each of the requested changes has been completed.

Round 2

Reviewer 1 Report

I appreciate the authors for the revision. The revised work addressed almost all my previous concerns. Some minor points to the revised manuscript need more attention.

Lines 354-356 need more editing work.

Lines 482-483, which two samples neutralize both SARS-CoV-2 and MERS-CoV pseudoviruses?

p.11, Figure 7 needs more work. The same color in each subplot represents a very different range of percentage identity, which is confusing. It would be better all these subplots use the same color scheme matched with the percentage similarity.

Lines 606-609, citation reference needed here.

Author Response

REVIEWER #1

I appreciate the authors for the revision. The revised work addressed almost all my previous concerns. Some minor points to the revised manuscript need more attention.

We thank the reviewer for the expert and constructive critique.

Lines 354-356 need more editing work.

Editing has been done.

Lines 482-483, which two samples neutralize both SARS-CoV-2 and MERS-CoV pseudoviruses?

This is noted in revision

p.11, Figure 7 needs more work. The same color in each subplot represents a very different range of percentage identity, which is confusing. It would be better all these subplots use the same color scheme matched with the percentage similarity.

We thank the reviewer for the suggestion, which has greatly improved the figure. The Fig. 7 subplots now use the same color scheme matched with the percentage similarity.

Lines 606-609, citation reference needed here.

We added the appropriate citation.

Reviewer 2 Report

The authors corrected the Manuscript following the suggestions of the reviewers or provided acceptable explanations. Furthermore, the authors pointed the limitations of the study which is very important for further proper interpretation of the presented results. 

Some typo editing has to be performed. For example:

1. line 91: “fever” has to be added after “Lassa”

2. line 397: “other” is duplicated

3. Line 539: “SARS-Cov-2” has to be replaced by “SARS-CoV-2”

4. Line 553: “bit” has to be replaced by “but”

5. Lines 596-598: The sentence has to be corrected (e.g. “Among Sierra Leonian blood samples tested in neutralization assays, approximately a quarter possessed a low titer of neutralizing antibodies against SARS-CoV-2 pseudovirus, while approximately a third had low titers of MERS-CoV neutralizing antibodies.”)

Author Response

REVIEWER #2

The authors corrected the Manuscript following the suggestions of the reviewers or provided acceptable explanations. Furthermore, the authors pointed the limitations of the study which is very important for further proper interpretation of the presented results. 

We thank the reviewer for the expert and constructive critique.

Some typo editing has to be performed. For example:

  1. line 91: “fever” has to be added after “Lassa”
  2. line 397: “other” is duplicated
  3. Line 539: “SARS-Cov-2” has to be replaced by “SARS-CoV-2”
  4. Line 553: “bit” has to be replaced by “but”
  5. Lines 596-598: The sentence has to be corrected (e.g. “Among Sierra Leonian blood samples tested in neutralization assays, approximately a quarter possessed a low titer of neutralizing antibodies against SARS-CoV-2 pseudovirus, while approximately a third had low titers of MERS-CoV neutralizing antibodies.”)

Each of the typos have been corrected and other editing has been completed.